# Dexamethasone-Loaded Radially Mesoporous Silica Nanoparticles for Sustained Anti-Inflammatory Effects in Rheumatoid Arthritis

**DOI:** 10.3390/pharmaceutics14050985

**Published:** 2022-05-04

**Authors:** Sang Jun Kim, Youngbo Choi, Khee Tae Min, Surin Hong

**Affiliations:** 1Seoul Jun Research Center, Seoul Jun Rehabilitation Clinic, Seoul 06737, Korea; catedral.sjk@gmail.com; 2Department of Safety Engineering, Chungbuk National University, Gyeongju 28644, Korea; ybc@cbnu.ac.kr; 3Department of Big Data, Chungbuk National University, Gyeongju 28644, Korea; 4Department of Biotechnology, CHA University, Seongnam 13488, Korea; undersnow93@naver.com

**Keywords:** anti-inflammatory, drug delivery, loading efficiency, rheumatoid arthritis, silica nanoparticles, sustained release

## Abstract

Radially mesoporous silica nanoparticles (RMSNs) with protonated amine functionality are proposed to be a dexamethasone (Dex) carrier that could achieve a sustained anti-inflammatory effect in rheumatoid arthritis (RA). High-capacity loading and a sustained release of target drugs were achieved by radially oriented mesopores and surface functionality. The maximum loading efficiency was confirmed to be about 76 wt%, which is about two times greater than that of representative mesopores silica, SBA-15. In addition, Dex-loaded RMSNs allow a sustained-release profile with about 92% of the loaded Dex for 100 h in vitro, resulting in 2.3-fold better delivery efficiency of Dex than that of the SBA-15 over the same period. In vivo evaluation of the inhibitory effects on inflammation in a RA disease rat model showed that, compared with the control groups, the group treated with Dex-loaded RMSNs sustained significant anti-inflammatory effects and recovery of cartilage over a period of 8 weeks. The in vivo effects were confirmed via micro-computed tomography, bone mineral density measurements, and modified Mankin scoring. The proposed Dex-loaded RMSNs prolonged the life of the in vivo concentrations of therapeutic agents and maximized their effect, which should encourage its application.

## 1. Introduction

Rheumatoid arthritis (RA) is an autoimmune disease that causes synovial inflammation and joint destruction accompanied by inflammation [1,2,3,4,5,6]. Conventional treatment focuses on the inhibition of inflammation. Among available therapeutic agents, a corticosteroid is a type of strong anti-inflammatory drug that inhibits inflammation and modifies the course of this disease [7,8,9]. Corticosteroid injections into joints stiffened by rheumatoid arthritis reduce synovial inflammation and decrease pain [10,11,12]. However, to achieve maximum anti-inflammatory treatment and prevent the systemic effect of the development of septic arthritis [13,14,15], it is important to achieve a prolonged concentration of corticosteroids in the synovial fluid and synovium [16].

Over the past few decades, a variety of approaches for efficient drug delivery systems have revealed targeted delivery and increased bioavailability, as well as solubilizing and improving the pharmacological profile of drugs [17,18,19,20]. In particular, the application of mesoporous silica materials as drug carriers has resulted in significant advancements in the field of drug delivery research [21]. These improvements include huge surface areas, large pore volumes, the ability to tailor the morphology and pore structure, controllable functionality of the surface, and good biocompatibility, which has made mesoporous silica materials prominent drug carriers [22,23,24,25,26,27]. Moreover, recent progress in synthetic methods has enabled the advent of new types of mesoporous silica nanoparticles that now feature distinctive pore structures and morphologies, which are expected to be highly useful as drug delivery systems.

For example, radially mesoporous silica nanoparticles (RMSNs) have attracted significant interest and have been explored in many applications [28,29,30]. The RMSNs have a radially wrinkled structure, and mesopores of the RMSNs are radially oriented. Furthermore, the mesopores of the RMSNs gradually widen from the center toward the outer surface of the RMSNs. The unique pore structure and morphology of RMSNs are expected to enable drug molecules to transfer readily into or out of the pores, which can be advantageous to loading capacity and to the release profiles of drugs. In our previous study, the RMSNs with protonated amine functionalities were able to load an anionic drug such as ibuprofen with loading efficiency as high as about 270 wt%, which is much greater than the reported data for conventional mesoporous silica materials, and the release profile of the loaded ibuprofen remained for 50 h in the in vitro test [31].

Merits such as high levels of drug-loading and long-lasting drug release make the RMSNs one of the most suitable candidates for a corticosteroid carrier with sustained anti-inflammatory effects in the RA. In order to investigate the loading efficiency, in vitro release profiles, and in vivo therapeutic effectiveness, careful experiments were designed as shown in Figure 1. In this study, we selected dexamethasone as an anti-inflammatory corticosteroid. Amine functionalities were introduced on the surface of the RMSNs by using a post-grafting method, and then the protonation of the amine functionality was induced under acidic conditions. Since a phosphate group in dexamethasone is negatively charged, the introduction of a protonated amine functionality to the RMSN was expected to further improve the loading efficiency by allowing charge–charge interactions between the RMSN and dexamethasone.

In addition, to compare the performance of the RMSN as a dexamethasone carrier with that of conventional mesoporous silica materials, loading capacities, and release profiles of dexamethasone-loaded RMSN and SBA-15, (i.e., Dex-RMSN and Dex-SBA-15, respectively), were examined under in vitro experimental conditions. Because SBA-15 is known to be a nontoxic material and a promising transporter for drugs, it has been extensively studied for use in drug delivery systems [32,33,34,35,36], and we selected SBA-15 as a representative of mesoporous silica materials for comparison. Moreover, to assess the prolonged anti-inflammatory effects of Dex-RMSN under in vivo conditions, RA model rats were prepared. The Dex-RMSNs were administrated into the knee joints of the RA model rats. Control groups were also prepared by administrating dexamethasone injections and saline injections into other RA model rats. The in vivo sustained effects were investigated via the time-resolved prognosis of cartilage destruction using micro-computed tomography (CT), bone mineral density (BMD) measurements, and modified Mankin scoring.

## 2. Materials and Methods

### 2.1. Materials

Tetraethyl orthosilicate (TEOS, 99%), cetylpyridinium bromide hydrate (CPB, 98%), cyclohexane (99%), urea, 3-aminopropyltriethoxysilane (3-APTES, 97%), anhydrous toluene (99.8%), poly(ethylene glycol)-block-poly(propylene glycol)-block-poly(ethylene glycol) (P123, M = 5800), and hydrochloric acid (HCl, 37 wt%) were purchased from Sigma-Aldrich Korea Ltd. The 1-Pentanol (99%) was received from Daejung Chemicals and Metals Co., Ltd. Monosodium iodoacetate (MIA, Bioworld, Cat No. 40950004, Dublin, OH, USA) was used to induce rheumatoid arthritis. Isoflurane gas (Ifran, 2%) was purchased from Hana Pharma Co., Seoul, Korea, and used for anesthetizing model rats. Clinical dexamethasone (Dex) solution (5 mg/mL in sodium phosphate, Yuhan Co., Seoul, Korea) was utilized for the loading experiments.

### 2.2. Preparation of Drug Carrier

The radially mesoporous silica nanoparticles (RMSNs) were synthesized using a reported method with minor modifications [31]. A total of 1.2 g of urea and 2 g of CPB were dissolved in 60 mL of distilled water. In addition, 5 g of TEOS and 3 mL of 1-pentanol were dissolved in 60 mL of cyclohexane. Then, the two solutions were mixed and stirred at room temperature for 30 min. The mixed solution was reacted at 120 °C for 4 h in an autoclave. After the reaction, RMSNs were collected from the mixed solution by centrifugation and washed with acetone, ethanol, and water. The washed RMSNs were sufficiently dried at room temperature and then calcined at 550 °C for 9 h.

SBA-15 was synthesized according to a previously established method [37,38,39]. P123 was added to a mixed solution that included 75 g of distilled water and 250 mL of 2 M HCl at 35 °C. The solution was stirred for 2 h after 21.5 g of TEOS was added. The solution was reacted at room temperature for 20 h and then heated at 100 °C for 24 h under static conditions. After the reaction, the resultant powder was collected by filtration, washed with distilled water, and calcined at 550 °C for 5 h.

The introduction of amine functionality on the RMSNs and SBA-15 was carried out using the post-grafting method [40,41]. One gram of RMSNs or SBA-15 was mixed with a solution involving 80 mL of anhydrous toluene and 4 mL of 3-APTES. The mixtures were then refluxed for 24 h and filtered. The obtained amine-functionalized RMSNs or SBA-15, (i.e., amine-RMSNs or amine-SBA-15, respectively), were washed with acetone and ethanol, followed by drying for 24 h at room temperature.

### 2.3. Characterization of Silica Carriers

Scanning electron microscopy (SEM) images were obtained using a Carl Zeiss SUPRA 55 VP field-emission scanning electron microscope. Transmission electron microscopy (TEM) observations were carried out using a JEOL JEM-3010 microscope operated at 300 kV. Fourier-transform infrared spectroscopy (FTIR) analysis was carried out using the SHIMADZU IRspirit-T model. The hydrodynamic size of RMSNs was measured using an ELSZ-2000ZS (Otsuka Electronics Co., Ltd., Osaka, Japan). The same instrument was also used to measure the zeta potentials of the RMSNs and SBA-15.

### 2.4. Preparation of Dexamethasone-Loaded Silica Carriers

0.1 g of the amine-RMSNs was mixed with 20 mL of solution including 10 mL of anhydrous ethyl alcohol and 10 mL of phosphate-buffered saline. In order to protonate the surface functionality of the RMSNs under different acidic conditions, HCl was used to adjust the pH values of the mixture to be 2, 2.5, 3, and 4. The mixture was then stirred at 500 rpm for 60 min, and the RMSNs with protonated amine functionality, (i.e., protonated amine-RMSNs) were recovered by centrifuging the mixture. A total of 100 mg of the Dex solution was dissolved into 20 mL of solution consisting of 10 mL of anhydrous ethyl alcohol and 10 mL of phosphate-buffered saline. The protonated amine-RMSNs were added to a beaker with the alcoholic mixture of the Dex solution and stirred at a constant rate of 500 rpm. From the mixture beaker, 1 mL samples of the solution were taken at intervals of 1 h for 4 h. To calculate the amount of Dex loaded into the RMSNs, the absorbance at 242 nm was measured using UV-visible spectroscopy. All loading experiments were conducted an average of 3 times. The loading efficiency was calculated using the following equation.
$$Loading\;efficiency\;\left(\%\right)=\frac{weight\;of\;dexamethasone\;in\;silica\;carrier}{weight\;of\;silica\;nanoparticles}\times 100\;$$

After loading experiments, dexamethasone-loaded RMSNs, (i.e., Dex-RMSNs) were retrieved via centrifugation at 12,000 rpm for 30 min, and then dried at room temperature. The fabrication of dexamethasone-loaded SBA-15, (i.e., Dex-SBA-15) and sample measurements were also conducted under the same conditions and methods.

### 2.5. In Vitro Dexamethasone Release from the Silica Carrier

A total of 0.1 g of Dex-RMSNs was dispersed in 100 mL of PBS buffer, and the concentration of released Dex from RMSNs into the release medium solution was monitored. During the experiments, the release medium solution was maintained at 36.5 °C using a water jacket, and the solution was stirred at a constant rate of 160 rpm. At specific time points, 1 mL of a sample was taken from the release medium solution, and 1 mL of fresh PBS buffer was added. The absorbance at 242 nm of the sample solution was measured using UV-visible spectroscopy, as were the loading experiments. The concentration of released Dex was calculated using the calibration curve, and the following equation was used to correct the concentration of released Dex in the medium solution.
$$C_{tcorr}=C_{t}+\frac{v}{V}\overset{t-1}{\underset{0}{\sum }}C_{t}$$

In this equation, *C_tcorr_* and *C_t_* are the corrected and apparent concentrations of released Dex at time *t*, respectively. *ν* is the volume of the sample taken from the release-medium solution, and *V* is the total volume of the release medium. Releasing profile of Dex was observed for 100 h. The releasing experiments of Dex from Dex-SBA-15 were also conducted under the same conditions and methods.

### 2.6. Preparation of Animal Models for Rheumatoid Arthritis

A total of 96 male Sprague-Dawley rats, 5 weeks of age, were used for this study. Following a 1-week acclimation period under a 12 h light/dark cycle with food and water ad libitum, the rats were anesthetized using 2% isoflurane gas. After removing the hair around both knee joints by clipper and disinfecting the knee joints with 70% ethanol, MIA solution was injected at both knee joints to establish a RA model according to a previously established method [42]. The MIA solution was prepared by dissolving 100 mg of MIA in 1 mL of sterile normal saline that was then filtered using 0.22 μm syringe filters. Animal experiments were reviewed and approved by the Institutional Animal Care and Use Committee (IACUC) of Samsung Biomedical Institute (SBRI) (Approval number 20150210002). SBRI is an Association for the Assessment and Accreditation of Laboratory Animal Care International (AAALAC International) accredited facility and abides by the guidelines set forth by the Institute of Laboratory Animal Resources (ILAR).

### 2.7. Evaluation of Anti-Inflammatory Effects In Vivo

Ten days after the MIA solution injection, the rats were divided into 3 groups: a Dex-injected group (*n* = 24), a Dex-loaded RMSNs (Dex-RMSNs)-injected group (*n* = 24), and a saline-injected group (*n* = 24). Dex (100 μL), Dex-RMSNs (100 μL), and 0.9% normal saline (100 μL) were administered to the left side of the knee joints in each group. After injections to each group, the rats in each group were euthanized for the evaluation of histology and micro-computed tomography (CT) at 1 week (*n* = 6), 2 weeks (*n* = 6), 4 weeks (*n* = 6), and 8 weeks (*n* = 6). To analyze the modified Mankin scores, an analysis variance (ANOVA) test was performed, and a Tukey test was used for post hoc analysis. A semi-quantitative method was introduced to grade the degree of arthritic changes used in a previous article [43]. Bone mineral density (BMD) was also measured at the femoral medial condyle area via repeated ANOVA testing. For the CT images, a scanning time of 0.21 s with settings of 80 kVp, 500 μA, and 30 calibrations was applied. Axial and transaxial fields of view of 30.74 mm were acquired.

For the histologic analysis, the knee joints of the rats were dissected and fixed in 10% neutral buffered formalin for 3 days. The fixed tissue was decalcified, embedded in paraffin, and cut to prepare sagittal sections. These sections were stained in hematoxylin and eosin (H & E) and Alcian blue method to evaluate the arthritic changes. The degree of degeneration was evaluated using a modified Mankin scoring system. Cartilage structure (0–6), chondrocytes (0–3), Alcian blue staining (0–4), and tidemark integrity (0–1) were evaluated as components [43].

### 2.8. Data Management and Statistical Analyses

Data were analyzed using descriptive statistical methods such as the mean ± standard deviation (SD) and multi-factor analysis of variance (ANOVA). A chi-square test was used to determine the change in medication between groups. SPPSS 20.0 software (IBM Corp., Chicago, IL, USA) was used for the analysis, and *p* values less than 0.01 (*p* < 0.001) were considered statistically significant.

## 3. Results and Discussion

### 3.1. Characterization of Silica Carriers

The representative SEM and TEM images featured in Figure 2 reveal the morphology and pore structures of the RMSNs, which agree well with the results of our previous work [31]. It is clear that the RMSNs have a radially wrinkled structure with a spherical shape. In detail, the RMSNs consist of wrinkled sheets that develop radially to a spherical form. Vacant spaces between the wrinkled sheets create pores, and thus the pores of the RMSNs are radially aligned. The TEM image in Figure 2B reveals that the size of the radially arranged pores widens gradually from the center toward the outer surface of the RMSNs. Due to these morphological features, the pores of the RMSNs have significantly wide entrances, as shown in the SEM image in Figure 2A. In addition, the size-distribution histogram in Figure 2C shows that RMSNs have uniform particle sizes with an average diameter of 513.6 ± 63.1 nm.

On the other hand, the TEM images in Figure 2D show the structural and morphological differences between the RMSNs and the most representative mesoporous silica, SBA-15. Highly ordered hexagonal pores appear in the TEM image of SBA-15, and the pores of SBA-15 have a relatively long length with a micrometer scale. Therefore, drug molecules must travel a relatively long distance in order to transfer into or out of the innermost parts of the pores of SBA-15, and this could easily be hindered by either the shrinkage or blockage of the hexagonal pores of SBA-15. However, the conical pore structures with a wide entrance in the RMSNs could enable the drug molecules to move readily into or out of the pores.

FTIR analysis of Figure 3 confirmed that the amine functionalization on the surface of the RMSNs and SBA-15 was successful. Compared with pristine RMSNs and SBA-15, the FTIR spectra for amine-RMSNs and SBA-15 clearly revealed peaks at around 2900 and 1550 cm*^−^*^1^, which correspond to the –CH_2_ and -NH bending groups in 3-APTES, respectively. All RMSNs and SBA-15 showed distinctive peaks at around 1050 and 810 cm*^−^*^1^, which correspond to the stretching and bending vibrations of Si–O–Si, respectively.

### 3.2. Dex Loading Efficiency of Silica Carriers

In order to achieve high loading efficiency for Dex drugs, it is necessary to induce specific and strong interactions between the surface of the silica carrier and Dex molecules. Functionalizing the surface of the silica carriers with an amine moiety could enhance the interaction with Dex molecules. Since the amine groups on the silica carriers have a pK_a_ value of about 9.26, they could have a positive charge even under neutral pH conditions and could interact electrostatically with a negatively charged phosphate moiety in Dex molecules. Zeta potential analysis supports these assumptions. Referring to Table 1, RMSNs and SBA-15 showed zeta potentials of −24.3 and −12.0 mV, respectively. After the amine-functionalization, the zeta potentials of the RMSNs and SBA-15 were increased to 12.6 and 3.2 mV, respectively. This indicates that the amine-functionalized surfaces of the amine–RMSNs and SBA-15 are positively charged.

We also performed protonation treatments on the amine-functionalized silica carriers under acidic conditions ranging between pH 2~4 to further enhance the interaction between the silica carriers and Dex molecules and promote a higher level of Dex loading capacity. Considering the equilibrium in the acid dissociation reaction between -NH_2_ and -NH_3_^+^ groups, pK_a_ is equal to the sum of the pH and log([-NH_3_^+^]/[-NH_2_]). Therefore, the ratio of -NH_3_^+^ to -NH_2_ groups could be increased at pH conditions lower than pK_a_ [44]. Moreover, silanol groups on the silica carrier could also be protonated under low pH conditions, which would increase the interaction with Dex molecules [45].

This speculation was in accordance with experimental results for the Dex loading efficiency. Figure 4 shows the UV-visible spectra of loading efficiency according to the pH treatment conditions of the protonated amine-RMSNs and SBA-15. As the protonation was carried out under conditions of lower pH, the protonated amine-RMSNs showed higher Dex-loading efficiency, and the highest Dex-loading efficiency of the protonated amine-RMSNs was achieved at pH 2. The Dex-loading efficiency of the protonated amine-SBA-15 was also varied with the pH conditions of the protonation in a way similar to the RMSNs. The protonated amine-RMSNs were more efficient at drug loading compared with the protonated amine-SBA-15. For the protonated amine-RMSNs, a maximum loading efficiency of about 76 wt% (≈0.076 g of Dex per 0.1 g of RMSNs) was established, which is twice that of the protonated amine- SBA-15 (about 38%).

The differences in zeta potential between the protonated amine-RMSNs and SBA-15 may contribute to a gap in the Dex loading efficiency. The protonated amine-RMSNs revealed zeta potential that was higher than protonated amine-SBA-15. This indicates that the density of positive charges with good accessibility is higher on the surface of the protonated amine-RMSNs than that of the protonated amine-SBA-15, and thus a greater amount of Dex molecules could interact with and be loaded onto the surface of the protonated amine-RMSNs. Considering that the protonated amine-RMSNs and SBA-15 consist of the same silica materials, and are treated by the same functionalizing and protonating process, their difference in zeta potential may be strongly affected by their pore structures. The radially originated mesopores with wide entrances may make the positively charged moieties in the pores of the protonated amine-RMSNs more accessible to Dex molecules, leading to higher zeta potential and higher loading efficiency.

### 3.3. In Vitro Investigation of the Release Profiles of Dex from Silica Carriers

In order to investigate the release profiles of Dex drugs under in vitro conditions, Dex release experiments were conducted in a release medium solution with a pH of 7.4. As shown in Figure 5, the Dex-RMSNs revealed a quite different release than Dex-SBA-15. In the early stage of the experiment, the RMSNs showed a rapid release profile, and about 80% of the loaded Dex was released from the RMSNs within 20 h. On the other hand, SBA-15 showed a relatively sustained release profile for about 40 h. These results for the initial release profiles may suggest that the SBA-15 could be a suitable Dex carrier for sustained therapeutic effects by comparison with the RMSNs.

However, if the release patterns at later stages of experiments are closely observed, it is clear that the RMSNs show promising positive results but the SBA-15 has significant limitations from the view of the substantially prolonged release of Dex. In the profile for Dex-RMSNs, the release of Dex did not stop even after the initial bursting release at about 20 h, and Dex was released continuously and steadily over a period of 80 h. Consequently, about 92% of the loaded Dex in the RMSNs was successfully released for 100 h. On the contrary, the release of Dex from SBA-15 was substantially ended after 40 h, and the release profile was not changed but reached a plateau for later 60 h. Therefore, the released amount of Dex from SBA-15 for 100 h was almost the same as that for the initial 40 h. In particular, only about 81% of the loaded Dex was released from SBA-15 for 100 h. Compared with the radially arranged conical mesopores of the RMSNs, the hexagonal and long mesopores of SBA-15 seemed to reduce the Dex releasing efficiency due to the relative difficulty in Dex releasing from deep inside of the pores. It should be noted that when the total Dex delivery amount was calculated quantitatively based on the loading efficiency per the same weight of silica carriers, the Dex delivery efficiency of the RMSNs over the long period of 100 h was about 2.3-fold better than that of the SBA-15. Therefore, the RMSNs demonstrated greater potential as drug carriers.

### 3.4. In Vivo Evaluation of Inhibition Effects of Inflammation in RA

Next, we moved to the in vivo evaluation for inhibition effects on inflammation in a RA disease rat model. RA-induced rats were treated with three different forms of drugs over several weeks, including clinical Dex formulation, Dex-RMSNs, and saline. The treatments were done by one-time injection of saline, Dex, and Dex-RMSNs at the same Dex dose of 0.1 mg/kg for all formulations in the three RA rat groups. The rats were observed daily for clinical symptoms, and femoral medial condyle areas were measured by micro CT every week. The 3D reconstructed images obtained after micro CT scanning of the patella of the femoral medial condyle area are represented in Figure 6. After rheumatoid arthritis induction, severe cartilage erosion was directly confirmed from the reconstructed image. The saline-injected group (Figure 6A(a),B(a)) demonstrated progressive cartilage erosion over 2 weeks. On the other hand, only one week after injection, we confirmed that cartilage erosion was significantly improved with Dex treatment (Figure 6A(b)) in the Dex-RMSNs-treated (Figure 6A(c)) group. Two weeks after injection, erosion of cartilage was observed in the Dex-only treated group (Figure 6B(b)), but the erosion of the cartilage had improved in the Dex-RMSNs-treated group (Figure 6B(c)). We inferred that the results were caused by the differences in the in vivo distribution of Dex concentrations according to the sustained release of Dex in the Dex-RMSNs-treated group, which reached a mineral-effective dose of the drug concentration and prolonged the inhibition of inflammation.

The histology images of knee joint samples shown in Figure 7 are consistent with the CT scanning data. In the saline-injected group, typical RA symptoms of the joint cavity gap, synovial hyperplasia, and fibrosis were observed, as shown in Figure 7A. The Dex-only and Dex-RMSNs-treated groups showed that the symptoms of joint cavity gap and fibrosis had been significantly improved in the Dex-RMSNs-treated group except for synovial hyperplasia symptoms (Figure 7B,C). Importantly, there were significant differences in improvement of the symptoms between the Dex-only and the Dex-RMSNs-treated groups, indicating that the Dex-RMSNs nanomedicine persistently inhibited and alleviated the effect of inflammatory infiltration in RA.

To investigate the alleviating effects, BMD was quantitated in the femur and tibia areas of each treated rat group. Figure 8A shows the BMD measurements of the femur and tibia for a total of 2 weeks in the saline, Dex-only, and Dex-RMSNs-treated groups. Total BMD was reported as an average value of measurements in these two areas. In the femur, the saline and Dex-only treated groups showed no significant differences in their BMD profiles between 1 and 2 weeks. However, we confirmed that the BMD scores increased in the Dex-RMSNs-treated group during the same period. In the tibia, the BMD scores had fallen in the saline and Dex-only treated groups over 2 weeks. On the other hand, the Dex-RMSNs-treated group maintained a similar level of BMD for 2 weeks. These results indicated that the Dex-RMSNs treatment was more helpful in maintaining bone mineral density, compared with the control group. The effects of the Dex-RMSNs treatment are more clearly shown in Figure 8B. In the saline and Dex-only treated groups, the BMD of the patella declined from the initial scores at 1 week over 8 weeks. However, the BMD of the patella was maintained through 8 weeks in the Dex-RMSNs-treated group, which was the result of the sustained release and longer-lasting effects of Dex in the body than that of the Dex-only treatment. Repeated measures of ANOVA testing supported the reliability of the data, and resulted in significant differences between the groups (F = 137.5, df = 3, *p* < 0.001) over time (F = 9.765, df = 7, *p* < 0.001).

Finally, the inhibitory effects on inflammation induced by drugs were evaluated using modified Mankin scoring. Figure 9 shows the mean values of histological grading from Mankin scores for the three different groups at different parts of the knee joint over time. In the saline-treated group, the Mankin scores gradually increased after 1 to 8 weeks, which was the result of the inflammatory effect. After only Dex treatment, the scores after 1 week were not significantly different from those of the saline-treated group (*p* = 0.079), and the scores decreased over 4 weeks, which was significantly different from the control group. Then, the scores greatly increased between 4 and 8 weeks at a similar level to the saline-treated group. The results showed that Dex delivered without carrier particle was not an efficient treatment for RA, which led to recurrent cartilage destruction and inflammation over time after treatment. On the other hand, in the Dex-RMSNs-treated group, the Mankin scores increased between 1 and 4 weeks, but the scores had significantly decreased from 4 to 8 weeks. At 8 weeks after treatment, the scores of the Dex-RMSNs-treated group, however, were significantly different from the control group (*p* < 0.001). The results showed that the Dex-RMSNs group retained the in vivo function of the therapeutic agents over time, and showed the best therapeutic status and prognosis of epiphysis destruction after 8 weeks, compared with the group treated only with Dex.

## 4. Conclusions

In conclusion, Dex-loaded RMSNs that feature high drug-loading efficiency and sustained release of drugs were proposed as a drug carrier for sustained anti-inflammatory effects in RA. The amine-functionalized RMSNs have radially wrinkled mesopores that induce charge–charge interactions with anionic drugs, which leads to high-capacity drug loading with a sustained release. The maximum loading efficiency into the RMSNs was attained at about 76 wt%, which is about twice that of SBA-15 under the same conditions. When the Dex-loaded RMSNs were applied to RA in a rat knee model, significantly better alleviative effects of inflammation were achieved compared with treatments for the control group. The effects were also confirmed through BMD measurement, and the recovery of cartilage destroyed at 8 weeks was confirmed through a modified Mankin score measurement. These findings suggest the potential for a nanomedicine that demonstrates biocompatible and controllable therapeutics in vivo.

## Figures and Tables

**Figure 1 pharmaceutics-14-00985-f001:**
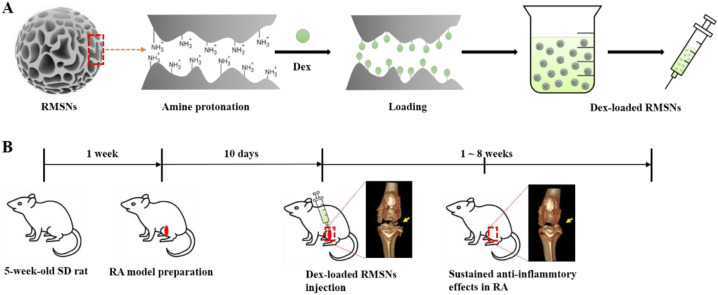
Schematic illustration of the steps involved in the preparation of Dex-loaded RMSNs (**A**); and, an experiment for sustained anti-inflammatory effects in an animal model of RA using Dex-loaded RMSNs (**B**).

**Figure 2 pharmaceutics-14-00985-f002:**
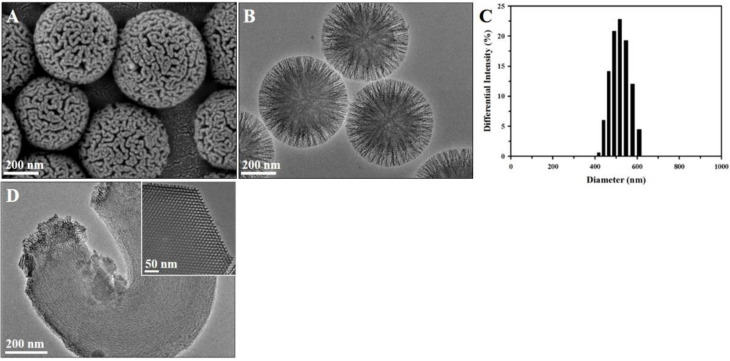
SEM images of the synthesized RMSNs (**A**); TEM images of the synthesized RMSNs (**B**); Size distribution histogram of RMSNs (**C**); and TEM images of the synthesized SBA-15 (**D**). The inserted TEM image shows the cross-section of an SBA-15.

**Figure 3 pharmaceutics-14-00985-f003:**
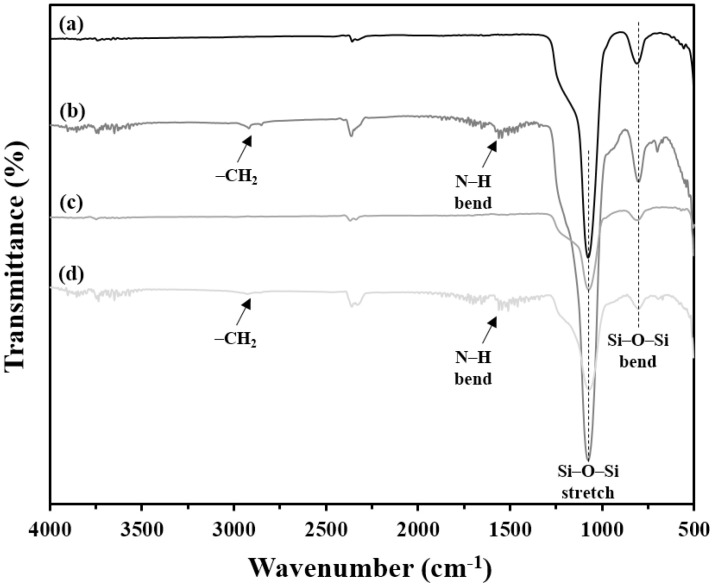
FTIR spectra of RMSNs (**a**), amine-RMSNs (**b**), SBA-15 (**c**), and amine-SBA-15 (**d**).

**Figure 4 pharmaceutics-14-00985-f004:**
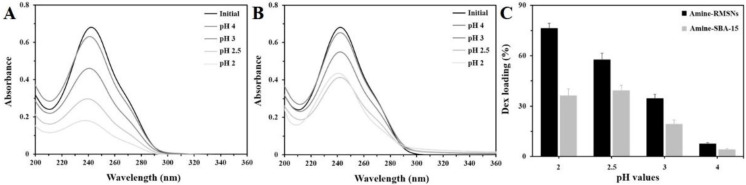
UV-Vis spectra for differences in the Dex loading efficiency according to pH treatment conditions on the surfaces of amine-RMSNs (**A**) and on amine-SBA-15 (**B**). Comparison of the Dex loading efficiency of amine-RMSNs with that of amine-SBA-15 (**C**) (*n* = 3, mean ± standard deviation).

**Figure 5 pharmaceutics-14-00985-f005:**
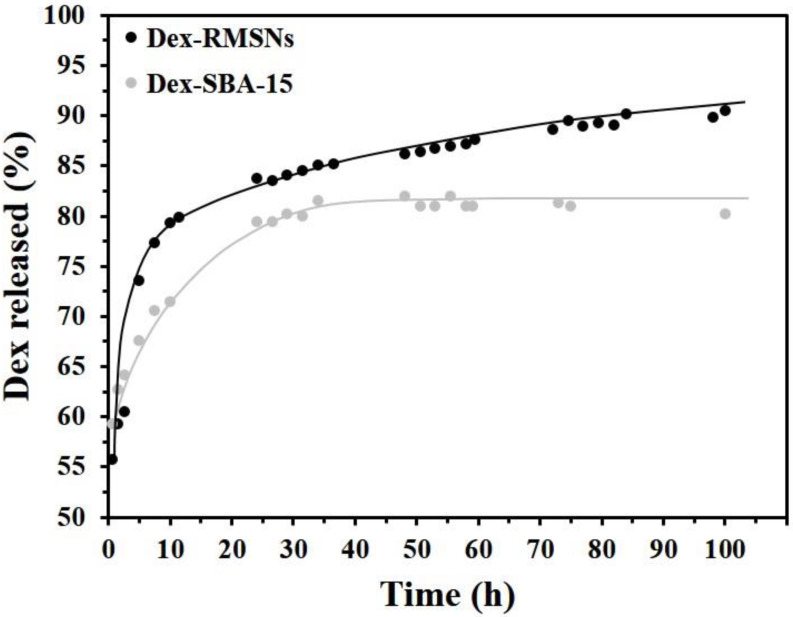
Time courses for release profiles of Dex from RMSNs and SBA-15.

**Figure 6 pharmaceutics-14-00985-f006:**
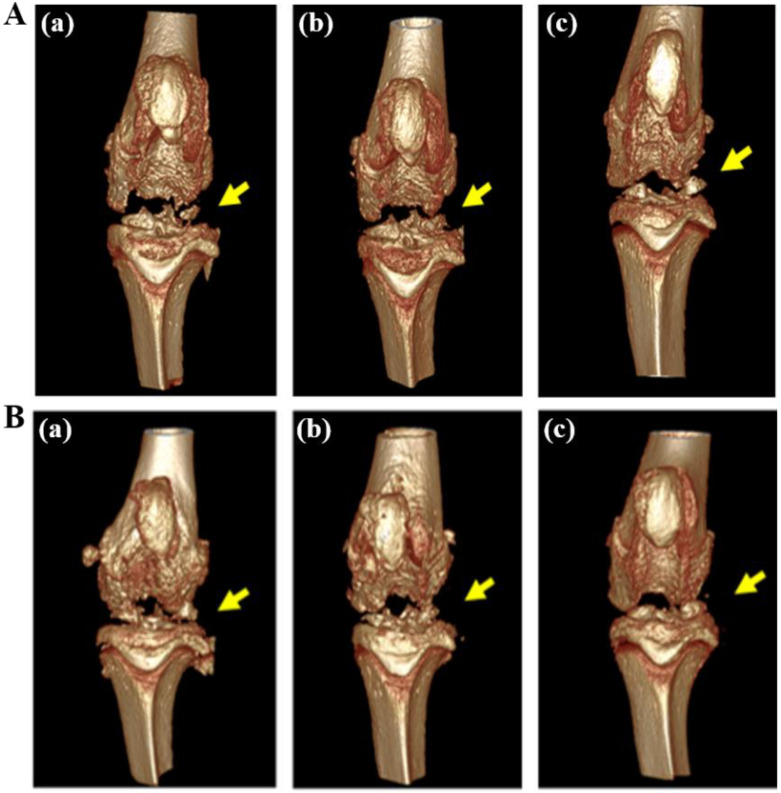
Micro-CT images of a rheumatoid arthritis model rat at 1 week (**A**) and at 2 weeks (**B**) after injection of saline (a), Dex (b), and Dex-RMSNs (c). Yellow arrow shows the place of erosion.

**Figure 7 pharmaceutics-14-00985-f007:**
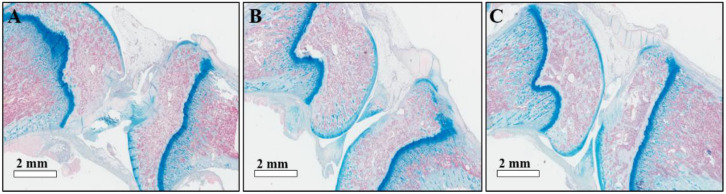
Histological images of knee joints treated with saline (**A**), Dex (**B**), and Dex-RMSNs (**C**). Tissues were stained with H & E and Alcian blue solution.

**Figure 8 pharmaceutics-14-00985-f008:**
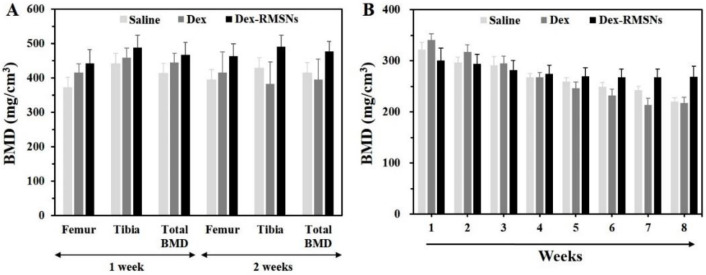
BMD of a femur and tibia after 1 week and 2 weeks in rheumatoid arthritis rats injected with saline, Dex, and Dex-RMSNs (**A**). BMD of the patella after 1 to 8 weeks in rheumatoid arthritis rats injected with saline, Dex, and Dex-RMSNs (**B**) (*n* = 6, mean ± standard deviation).

**Figure 9 pharmaceutics-14-00985-f009:**
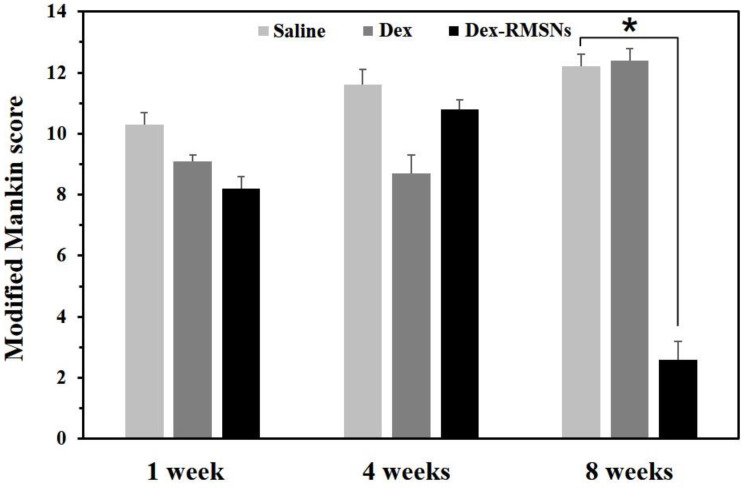
Modified Mankin scores in different parts of knee joints after treatments of saline, Dex only, and Dex-RMSNs at 1, 4, and 8 weeks. The error bar refers to the standard deviation (*n* = 6, mean ± standard deviation, * *p* < 0.001).

**Table 1 pharmaceutics-14-00985-t001:** Zeta-potential of different RMSNs and SBA-15 samples.

Sample	Zeta-Potential (mV)
RMSNs	−24.3 ± 1.0
Amine–RMSNs	12.6 ± 1.8
Protonated amine-RMSNs	56.0 ± 2.5
SBA-15	−12.0 ± 1.4
Amine-SBA-15	3.2 ± 3.2
Protonated amine-SBA-15	35.6 ± 2.6

## Data Availability

Data are contained within the article.

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
