# Peer review of "Dexamethasone-Loaded Radially Mesoporous Silica Nanoparticles for Sustained Anti-Inflammatory Effects in Rheumatoid Arthritis"

_pharmaceutics, 2022, doi:10.3390/pharmaceutics14050985_

Round 1

Reviewer 1 Report

In this paper, the authors loaded radially mesoporous silica nanoparticles with dexamethasone to treat rheumatoid arthritis. The paper is interesting and deserves publication following minor revisions.

The major concern is related to the fact that the MDPI template has not been used. The authors should use it.

Lines 10 and 11: 5 and 6 are not affiliations. 5 has to be substituted with an asterisk and 6 with a different symbol.

Abstract

All the acronyms have to be defined when used for the first time. Therefore, RA, CT and BMD have to be defined before using them in the abstract.

Introduction

In line 40, the authors define “corticosteroid” as a drug. It is not correct. Corticosteroids are a category of drugs and not a single compound.

Lines 46-48: Many innovative techniques have been employed to increase the bioavailability of active principles, such as the ones based on the use of supercritical fluids. See, for example, doi: 10.1016/j.cej.2016.02.041; doi: 10.1016/S0378-5173(01)00870-5.

Materials and methods

This section has been properly conducted and the methodologies accurately defined.

Results and discussion

The results have been clearly presented and the figures are explanatory; they deeply help the reader in the comprehension of the text.

Conclusions

The conclusions are supported by data.

Reviewer 2 Report

The manuscript entitled" Dexamethasone-loaded radially mesoporous silica nanoparticles for sustained anti-inflammatory effects in rheumatoid arthritis" by Kim et al, describes the anti-inflammatory effect of the Dex-loaded RMSNs in comparison with only Dex. The authors performed detailed characterization and efficacy evaluation with sufficient literature. Overall, their presentation is excellent, and findings are very interesting. However, as we know silica nanoparticles are toxic, how the authors overcome these issues in moving forward?

  1. what is the RMSN: DEX ratio?
